# Molecular Simulation of Improved Mechanical Properties and Thermal Stability of Insulation Paper Cellulose by Modification with Silane-Coupling-Agent-Grafted Nano-SiO$_2$

**Zhengxiang Zhang [1], Haibin Zhou [1], Wentao Li [1] and Chao Tang [2,\*]**

[1] Extra-High Voltage Transmission Company, China Southern Power Grid, Guangzhou 510663, China; Zhangzhengxiang@mail.ehv.csg.cn (Z.Z.); zhouhaibin@mail.ehv.csg.cn (H.Z.); liwenta@mail.ehv.csg.cn (W.L.)

[2] College of Engineering and Technology, Southwest University, Chongqing 400715, China

\* Correspondence: swutc@swu.edu.cn; Tel./Fax: +86-023-68251265

**Abstract:** Cellulose is an important part of transformer insulation paper. Thermal aging of cellulose occurs in long-term operation of transformers, which deteriorates the mechanical properties and thermal stability of cellulose, resulting in a decrease in the transformer life. Therefore, improvement of the mechanical properties and thermal stability of cellulose has become a research hotspot. In this study, the effects of different silane coupling agents on the mechanical properties and thermal stability of modified cellulose were studied. The simulation results showed that the mechanical parameters of cellulose are only slightly improved by KH560 ($\gamma$-glycidyl ether oxypropyl trimethoxysilane) and KH570 ($\gamma$-methylacrylloxy propyl trimethoxy silane) modified nano-SiO$_2$, while the mechanical parameters of cellulose are greatly improved by KH550 ($\gamma$-aminopropyl triethoxy silane) and KH792 (*N*-(2-aminoethyl)-3-amino propyl trimethoxy silane) modified nano-SiO$_2$. The glass-transition temperature of the composite model is 24 K higher than that of the unmodified model. The mechanism of the change of the glass-transition temperature was analyzed from the point of view of free-volume theory. The main reason for the change of the glass-transition temperature is that the free volume abruptly changes, which increases the space for movement of the cellulose chain and accelerates the whole movement of the molecular chain. Therefore, modifying cellulose with KH792-modified nano-SiO$_2$ can significantly enhance the thermal stability of cellulose.

**Keywords:** nano-SiO$_2$; silane coupling agent; thermal stability; mechanical parameter; molecular simulation





## 1. Introduction

Oil–paper insulation systems guarantee safe operation of oil-immersed transformers, but they are affected by various factors in long-term transformer operation [1–5]. Improvement of the mechanical properties and thermal stability of cellulose has become a research hotspot. In recent years, the rise of nanotechnology has prompted research on nanoparticle-modified cellulose insulating paper. Many researchers have made progress in this field [6–10]. Liu used nano-Al$_2$O$_3$ to modify insulating paper and studied the depth and degree of the traps [11]. The results showed that with increasing trap depth and density, the breakdown field strength of the modified insulating paper increases, and the conductivity decreases. The trap characteristics are the main reasons for the change of the dielectric properties of the modified insulating paper. Liao used nanoparticles to modify cellulose insulating paper, added SiO$_2$ hollow microspheres and soaked the modified insulating paper, and tested and analyzed its dielectric constant and breakdown voltage [12]. When the content of SiO$_2$ hollow microspheres was 5%, the dielectric constant of the modified insulating paper was 34% lower than that of the unmodified insulating paper, and the breakdown voltage of the oil–paper composite insulation system increased by 15.5%. However, nanoparticles agglomerate in the process of modification. At present, the main method to avoid agglomeration of nanoparticles is to modify the nanoparticles

with silane coupling agents. Wang investigated the influences of different types of silane coupling agents and grafting rate on the interface between nano-SiO$_2$ and cellulose [13]. They found that the interface effect was different for silane coupling agents with different chain lengths and groups. Therefore, it is necessary to study the mechanical properties and thermal stability of cellulose modified by different types of silane coupling agents.

In this study, the nano-SiO$_2$ content in all of the models was 5% [9,12,14]. The mechanical properties (tensile modulus, shear modulus, volume model, and Poisson's ratio) and thermal stability (glass-transition temperature and free-volume fraction) of cellulose modified by nano-SiO$_2$ modified by different silane coupling agents were investigated and compared through a molecular dynamics simulation based on the same grafting rate of the silane coupling agent. The mechanism of the change of the properties and thermal stability was also investigated.

## 2. Molecular Dynamics Simulation

Because of the high surface activity of nano-SiO$_2$, it is easily oxidized and produces hydroxyl groups on the nano-SiO$_2$ surface. First, the surface of the nano-SiO$_2$ model needs to be hydroxylated (the unsaturated O atoms on the surface must be treated with H and the unsaturated Si atoms with -OH). Second, the hydroxyl groups at one end of the silane coupling agent combine with the hydroxyl groups on the surface of nano-SiO$_2$ to form Si–O–Si so that the silane coupling agent can randomly connect to the O atoms on the surface of nano-SiO$_2$. There are three hydroxyl groups in a general silane coupling agent. One of the hydroxyl groups of the silane coupling agent bonds to a hydroxyl group on the surface of nano-SiO$_2$, and the remaining two hydroxyl groups form condensation with other grafted silane coupling agent molecules or exist in a free state. To simplify the model, we considered that the other two hydroxyl groups in the model do not participate in the condensation reaction.

Wang et al. [13] found that the grafting ratio is relatively good when the grafting quantity is 4, and the influence of the size of the model on the simulation time has also been investigated [15–17]. Therefore, different types of silane coupling agents with grafting quantity of 4 were grafted on the surface of nano-SiO$_2$, and the particle size of nano-SiO$_2$ was 5 Å. First, nano-SiO$_2$ was placed in a cell with dimensions of 40 × 40 × 40 Å, and its centroid was set to coincide with the centroid of the cell. The cellulose chain (degree of polymerization = 10) was filled in [18] using the packing function in the AC module, and the initial density was set to 0.6 g/cm$^3$. By applying pressure to different models, the density can reach 1.45 g/cm$^3$. Subsequently, the relevant parameters were calculated and analyzed. The composite models constructed in this study are shown in Figure 1.

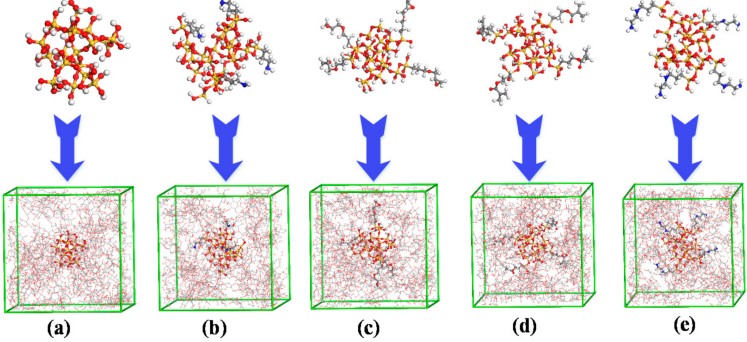

**Figure 1.** Composite models: (**a**) unmodified model, (**b**) KH550 model, (**c**) KH560 model, (**d**) KH570 model, and (**e**) KH792 model.

In this study, we used Molecular Simulation (MS) software. First, 5000 steps of geometric optimization were performed for the established amorphous model of nano-SiO$_2$ and cellulose. Second, annealing of the model was performed with annealing cycle of 5, initial temperature of 300 K, and heating ramps per cycle of 13. Finally, 100 steps of the annealed

convergence model were performed in the NVT ensemble. The dynamic simulation of 200 ps was performed in the NPT ensemble, and then the rationality criterion of the model after the dynamic simulation was obtained. After the system reached a stable configuration, the relevant parameters were analyzed. The pressure was set to 0.0001 GPa (standard atmospheric pressure), the Ewald method was used for the electrostatic interactions, the atom-based method was used for the van der Waals interactions, and the COMPASS force field was used. A Nose–Hoover thermostat was used to collect the dynamic information every 0.5 ps [19,20].

## 3. Results and Discussion

### 3.1. Mechanical Properties

The mechanical properties of materials are important indicators for characterizing the mechanical strength. In this study, the relevant mechanical parameters of cellulose were obtained by static elastic constant analysis (Table 1). The tensile modulus ($E$) is the ratio of the stress to the strain. The material has higher rigidity and greater ability to resist deformation due to the larger $E$. The shear modulus ($G$) is the ratio of the shear stress to the strain. The bulk modulus ($K$) is the incompressibility of the material. Poisson's ratio ($V$) is the plasticity of the material, with a high value indicating high plasticity [21,22]. The mechanical parameters of the models are shown in Figure 2. With increasing temperature, the modulus values ($E$, $G$, and $K$) of each model gradually decrease, while Poisson's ratio ($V$) is relatively stable. This indicates that the temperature has a relatively large effect on the deformation resistance, shear deformation resistance, and incompressibility of cellulose, while the effect of temperature on the plasticity of cellulose is relatively small. Comparing the mechanical properties of the models with four commonly used silane coupling agents, the mechanical parameters ($E$, $K$, and $V$) of cellulose are only slightly improved by KH560- and KH570-modified nano-$SiO_2$, but the mechanical parameters ($E$ and $K$) of cellulose are greatly improved by KH550- and KH792-modified nano-$SiO_2$, although Poisson's ratio ($V$) is not significantly improved. Therefore, compared with KH560 and KH570, nano-$SiO_2$ modified with silane coupling agents containing amino groups (KH550 and KH792) can significantly improve the deformation resistance, shear deformation resistance, and incompressibility of cellulose, with little effect on the plasticity.

**Table 1.** Free volume of the unmodified nano-$SiO_2$ and cellulose composite model.

| Temperature | Free Volume (A3) | Occupied Volume (A3) | FFV |
|---|---|---|---|
| 303 K | 7631.24 | 54,868.76 | 0.139082 |
| 323 K | 7701.92 | 54,398.08 | 0.141584 |
| 343 K | 7744.04 | 54,615.96 | 0.141791 |
| 363 K | 7739.88 | 54,460.12 | 0.14212 |
| 383 K | 7784.60 | 54,315.4 | 0.143322 |
| 403 K | 7833.52 | 54,466.48 | 0.143823 |
| 423 K | 8251.68 | 54,760.64 | 0.150686 |
| 443 K | 8332.32 | 54,267.68 | 0.153541 |
| 463 K | 8371.68 | 54,428.32 | 0.153811 |
| 483 K | 8436.20 | 54,863.80 | 0.153766 |
| 503 K | 8631.24 | 54,868.76 | 0.157307 |

### 3.2. Cross Energy

The energy is the most important property for the stability of the system. The energy of the system can be expressed as

$$E_{total} = (E_{internal} + E_{cross}) + E_{nonbond} \tag{1}$$

where $E_{\text{internal}} + E_{\text{cross}}$ is the bond interaction term and $E_{nonbond}$ is the non-bonding interaction term. The changes of the overall potential energy and non-bonding interaction energy of the models with temperature are shown in Figure 3.

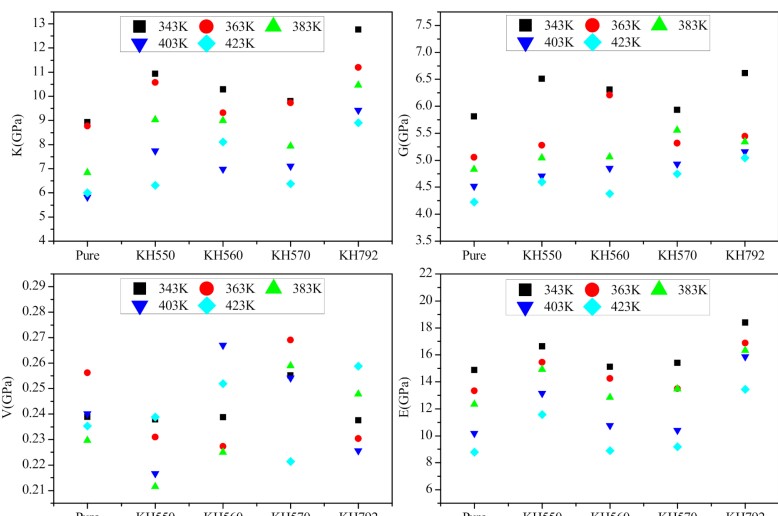

**Figure 2.** Mechanical parameters of the models.

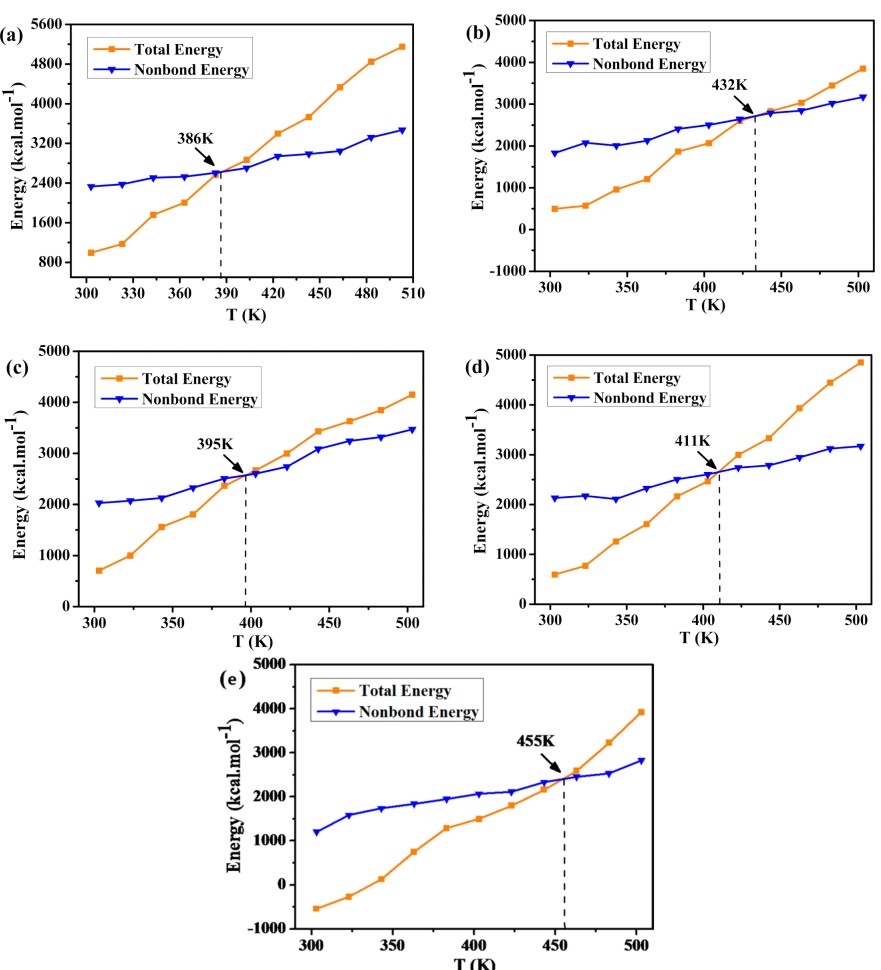

**Figure 3.** Energies of the models: (**a**) ungrafted model, (**b**) KH550 model, (**c**) KH560 model, (**d**) KH570 model, and (**e**) KH792 model.

From Figure 3, the total potential energy and non-bonding interaction energy of all of the models essentially linearly increase, but the growth rate of the non-bonding interaction energy is significantly lower than that of the total potential energy. In the ungrafted model and KH550, KH560, KH570, and KH792 composite models, the starting temperatures of the total potential energies are 386, 395, 411, 432, and 455 K, respectively. From Equation (1), when the total potential energy of the system becomes greater than the non-bonding interaction energy, the bond energy interaction term in the system changes from the previously negative value to a positive value, while the repulsion force between the atoms is greater than the attraction force between the atoms in the molecule, and the chemical bonds become unstable. When the temperature gradually increases, some active bonds and glycoside bonds are easy to break, which is the phenomenon of cellulose thermal degradation. Once cellulose thermal degradation occurs, the degree of polymerization of cellulose decreases, which leads to deterioration of the mechanical properties of cellulose. Therefore, the main reason for the decrease of the mechanical properties of cellulose is that the internal atomic force changes from attractive to repulsive, which makes the chemical bonds unstable. The order of the temperatures corresponding to the cross-energy points of the different models is KH792 > KH550 > KH560 > KH570 > pure.

### 3.3. Glass-Transition Temperature

The glass-transition temperature is the temperature of the transition from the glass state to the high-elastic state. As a glassy substance, cellulose has a glassy state, a high-elastic state, and a viscous-flow state, which are the three states of the polymer related to temperature. The transition points between these states are called the glass-transition temperature and melting point. The thermodynamic properties of the material obviously change before and after the glass-transition temperature. Therefore, it is of great importance to study the glass-transition characteristics of cellulose at high temperature to improve its thermal stability. When studying the glass-transition temperatures of polymers, the most commonly used and reliable method is the specific volume–temperature curve method [23–26]. The specific volume is the reciprocal of the density, and it is the most commonly used physical quantity in determination of the glass-transition temperature. During the glass transition, the specific volume and other properties of the material significantly change along with molecular segment movement, and the curve of the specific volume against the temperature abruptly changes. In this method, the specific volume obtained by the NPT molecular dynamics simulation is plotted against the temperature. The inflection point of the specific volume change with temperature is determined, and linear fitting is performed before and after the inflection point. The intersection of the two fitted curves is the turning point of vitrification, and the corresponding temperature is the vitrification temperature. The obtained fitted curves of the vitrification temperature for the unmodified and KH792-modified models are shown in Figure 4.

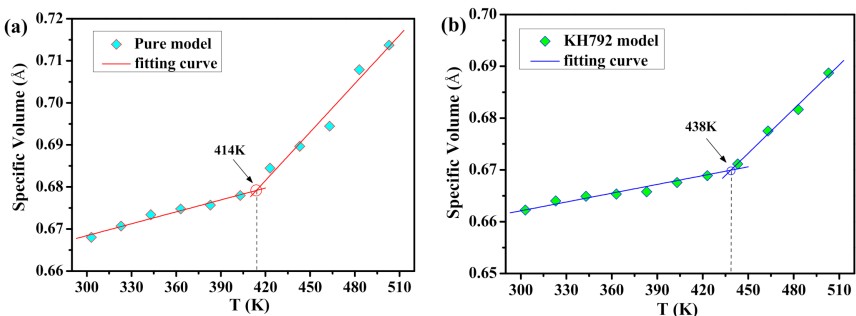

**Figure 4.** Specific volume–temperature curves of the models: (**a**) unmodified model and (**b**) KH792 model.

From Figure 4, the glass-transition temperature of the cellulose and unmodified nano-SiO$_2$ composite model is 414 K, while that of the KH792-modified nano-SiO$_2$ and

cellulose composite model is 438 K. Thus, the glass-transition temperature of the KH792-modified nano-SiO$_2$ and cellulose composite model increases by 24 K. The glass-transition temperature range of cellulose in the polymer manual is 250 to 580 K, so the value of the glass-transition temperature obtained from simulation analysis in this study is reasonable. However, it is only for reference, and it does not affect comparison between the glass-transition temperatures of the modified model and unmodified model. The glass-transition temperature of the KH792-modified nano-SiO$_2$ and cellulose composite model is higher than that of the unmodified nano-SiO$_2$ and cellulose composite model, which shows that the glass state to high-elastic state transition of the cellulose chain modified by KH792 is slower than that of the unmodified model, which effectively delays the cellulose chain transition from the glass state to the high-elastic state. Therefore, compared with the unmodified composite model, for the cellulose chain modified by KH792, the modified nano-SiO$_2$ and cellulose composite model can maintain better performance, the thermodynamic performance is less affected by temperature, and the thermal stability is significantly enhanced.

### 3.4. Free-Volume Fraction

In the molecular dynamics simulation, the free volume can be determined by the interstices between the chains in the model system. The free volume and occupied volume of the system can be measured by the Atom Volumes and Surface Tool in MS, and the free-volume fraction (FFV) of the system can be obtained by the ratio of the free volume to the total volume of the system (the sum of the free volume and occupied volume). Because of the different expansion coefficient, the occupied volume of the polymer linearly increases with temperature, and the free volume abruptly increases near the glass-transition temperature. Therefore, the mechanism of the glass-transition temperature was analyzed using free-volume theory. When calculating the free volume, the hard ball probe method was used, and the probe radius was set to 1 Å [26–28]. The free-volume distributions of the unmodified composite model and KH792-modified composite model are shown in Figure 5. The blue parts of the figures are the free-volume distribution areas. The free volumes of the unmodified composite model and KH792-modified composite model are given in Tables 1 and 2, respectively.

**Table 2.** Free volume of the KH792-modified nano-SiO$_2$ and cellulose composite model.

| Temperature | Free Volume (A3) | Occupied Volume (A3) | FFV |
|:---:|:---:|:---:|:---:|
| 303 K | 5424.80 | 56,672.80 | 0.087359 |
| 323 K | 5422.48 | 57,077.52 | 0.086760 |
| 343 K | 5492.64 | 56,907.36 | 0.088023 |
| 363 K | 5543.68 | 56,756.32 | 0.088984 |
| 383 K | 5624.60 | 56,532.08 | 0.090491 |
| 403 K | 5667.92 | 56,775.40 | 0.090769 |
| 423 K | 5687.43 | 56,301.52 | 0.091749 |
| 443 K | 6198.48 | 56,689.52 | 0.098564 |
| 463 K | 6147.60 | 56,352.40 | 0.098362 |
| 483 K | 6309.80 | 56,190.20 | 0.100957 |
| 503 K | 6511.24 | 56,908.76 | 0.102669 |

From Table 1, for the unmodified nano-SiO$_2$ composite model, when the temperature is 303–403 K, the change of the free volume is relatively small, However, above 423 K, the free volume starts to increase, and a clear increasing trend appears. From Table 2, for the KH792-modified nano-SiO$_2$ composite model, when the temperature is 303–423 K, the change of the free volume is small. However, above 443 K, the free volume starts to increase, and the free volume also shows a significant increasing trend. The sudden increase of the free volume increases the space for movement of the cellulose chain, leading to increased movement. The cellulose changes from the glassy state to the high-elastic state, and the thermal stability of the cellulose chain sharply decreases after it changes to the

high-elastic state. The free volume increase of the unmodified nano-SiO$_2$ composite model occurs at 403–423 K, which is essentially the same as the glass-transition temperature of the unmodified nano-SiO$_2$ composite model obtained by the specific-volume method in Section 3.3 (413 K). The free volume increase of the KH792-modified nano-SiO$_2$ composite model occurs at 423–443 K. This is consistent with the glass-transition temperature of the modified composite model obtained by the specific-volume method in Section 3.3 (438 K). Therefore, this section explains the microscale mechanism of improvement of the thermal stability of cellulose modified by KH792-modified nano-SiO$_2$ from the perspective of the free volume.

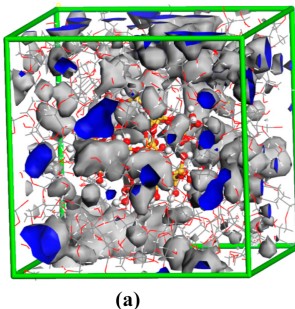 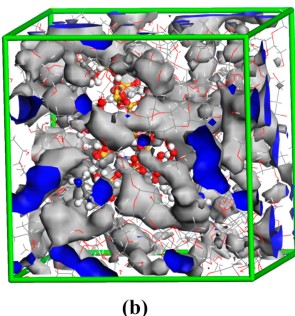

(**a**)  (**b**)

**Figure 5.** Free volume of the models: (**a**) unmodified model and (**b**) KH792 model.

Above all, hydroxyl groups are produced after hydrolysis of silane coupling agent. Therefore, the silane coupling agent can react with the silicon hydroxyl groups on the surface of nano-SiO$_2$, so that one end of the silane coupling agent can be connected with the surface of nano-SiO$_2$. The organic functional group of KH550 is amino, the organic functional group of KH560 is epoxy, the organic functional group of KH570 is acyloxy, and the organic functional group of kh792 is aminopropyl. Organic functional groups determine the binding ability between nano-SiO$_2$ modified by silane coupling agent and cellulose. The addition of silane coupling agent enhances the compatibility between organic phase and inorganic phase to a certain extent. At the same time, the addition of silane coupling agent creates more hydrogen bonds between nano-SiO$_2$ and cellulose, which makes the structure of cellulose more compact.

## 4. Conclusions

In this study, the effects of silane coupling agents (KH550, KH560, KH570, and KH792) on the mechanical properties and thermal stability of cellulose were investigated with molecular dynamics simulation. It was found that the mechanical properties and thermal stability of cellulose can be improved by silane coupling agents.

The temperature has a great effect on the deformation resistance, shear deformation resistance, and incompressibility of cellulose, but it has little effect on its plasticity. The mechanical parameters (*E*, *K*, and *V*) of cellulose are only slightly improved by KH560- and KH570-modified nano-SiO$_2$. However, the mechanical parameters (*E* and *K*) of cellulose are significantly improved by KH550- and KH792-modified nano-SiO$_2$, while Poisson's ratio (*V*) is only slightly improved. Therefore, this shows that silane-coupling-agent-modified nano-SiO$_2$ mainly affects the deformation resistance, shear deformation resistance, and incompressibility of cellulose, but it has little effect on its plasticity. Silane coupling agents containing amino groups significantly improve the deformation resistance, shear deformation resistance, and incompressibility of cellulose. The mechanism of the change of the mechanical properties was analyzed from the point of view of the energy. The reason for the decrease of the mechanical properties of cellulose is that the internal atomic force changes from attractive to repulsive, which makes the chemical bonds unstable. The overall potential energy obtained from energy analysis is higher than the starting temperature of the non-bonding energy, and the analysis results of the mechanical properties are essentially the same. Therefore, nano-SiO$_2$ modified by silane coupling agents can enhance

the mechanical properties of cellulose, and nano-$SiO_2$ modified by KH792 most significantly improves the mechanical properties of cellulose.

The glass-transition temperature of the nano-$SiO_2$-modified cellulose composite model without surface modification is 414 K, and that of the KH792-modified nano-$SiO_2$ composite model is 438 K. Thus, the glass-transition temperature of the composite model increases by 24% after surface modification of nano-$SiO_2$ by KH792. The mechanism of the change of the glass-transition temperature was analyzed from the point of view of free-volume theory. The main reason for the change of the glass-transition temperature is that the free volume abruptly changes, which increases the space for cellulose chain movement and intensifies the overall movement of the molecular chain. The range of the glass-transition temperature obtained by free-volume theory is essentially the same as that obtained by the specific-volume method. Therefore, the thermal stability of cellulose is significantly enhanced by modifying cellulose by KH792-modified nano-$SiO_2$.

**Author Contributions:** Conceptualization, Z.Z. and C.T.; formal analysis, W.L. and C.T.; software, H.Z.; validation, Z.Z., H.Z. and C.T.; data curation, Z.Z., H.Z. and C.T. All authors have read and agreed to the published version of the manuscript.

**Funding:** This research was funded by the science and technology project of EHV Power Transmission Company, China Southern Power Grid.

**Institutional Review Board Statement:** Not applicable.

**Informed Consent Statement:** Not applicable.

**Data Availability Statement:** The data that support the findings of this study are available from the corresponding author, C.T., upon reasonable request.

**Conflicts of Interest:** The funders had no role in the design of the study; in the collection, analyses, or interpretation of data; in the writing of the manuscript, or in the decision to publish the results.

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
