# Peer review of "Molecular Simulation of Improved Mechanical Properties and Thermal Stability of Insulation Paper Cellulose by Modification with Silane-Coupling-Agent-Grafted Nano-SiO2"

_processes, doi:10.3390/pr9050766_

Round 1

Reviewer 1 Report

The paper refers to improving thermal stability and mechanical properties of insulation paper. Although the topic is interesting and up-to-date in present form the paper is unacceptable for publication. Essential information is missing. What software was used? The authors used for simulations coupling agents with acronyms: KH550, KH560, KH570 and KH792 without description of these substances. The results are speculative. The work has modelling character, no experimental part has been carried out, thus conclusions as well as e.g. the sentence from abstract: “… the mechanical properties are greatly improved by KH550…” has no confirmation. Moreover, the title misleads suggesting the experimental character of work.

English should be reedited thoroughly. Many typographic errors appear, e.g. “Nano-SiO2”, including the title. The language lacks a scientific nomenclature, e.g.:

- “we guarantee that the content of Nano-SiO2 in all models is 5%”

-p.2 l.62-63, l.66, l.75: “Due to high surface activity of nano-SiO2 there will be a lot of OH groups on the surface” (additionally, SiO2 has no OH group), and: “…agent will shrink with the hydroxyl on the surface…”, and: “…coupling agents with the amount of 4.”

Free volume fraction is not a thermal property, as well as Tg is not “thermal stability” parameter. These parameters correlate with mobility of the molecules and the same Tg, however to determine thermal stability TGA is recommended.

TG is not a symbol for glass transition temperature, it is for “thermal gravimetry”.

Moreover, the references should be supplemented, as only the names of the two first authors were given.

Author Response

(1)What software was used?

The software used in this paper is Molecular Simulation (MS), we’ve added in the revised manuscript.

(2)The authors used for simulations coupling agents with acronyms: KH550, KH560, KH570 and KH792 without description of these substances.

  Thanks for your comments.

We supplement the descriptions of KH550, KH560, KH570 and KH792 and add them to the revised manuscript. KH550 (γ-aminopropyl triethoxy silane),KH560 (γ-glycidyl ether oxypropyl trimethoxysilane),KH570 (γ-methylacrylloxy propyl trimethoxy silane),KH792 (N-(2-aminoethyl)-3-amino propyl trimethoxy silane).

(3)The work has modelling character, no experimental part has been carried out, thus conclusions as well as e.g. the sentence from abstract: “… the mechanical properties are greatly improved by KH550…” has no confirmation. Moreover, the title misleads suggesting the experimental character of work.

Thanks for your comments.

In this paper, the improvement of cellulose mechanical properties by silane coupling agent modified nano silica was studied from the micro level. We modified the title and highlighted the theme of molecular simulation.

(4)English should be reedited thoroughly

Thanks for your comments.

We invited a professional English native speaker from Edanz group to retouch the manuscript and check the language and spelling.

(5)additionally, SiO2 has no OH group

Thanks for your comments.

Silica itself is not OH, but because of its surface instability, it will be oh treated in the production process. So in the process of simulation modeling, hydroxylation will be carried out generally.

(6)Free volume fraction is not a thermal property, as well as Tg is not “thermal stability” parameter. These parameters correlate with mobility of the molecules and the same Tg, however to determine thermal stability TGA is recommended.

Thanks for your comments.

TGA can really improve the thermal properties of the reaction materials, but this paper is from the micro level, and has not carried out relevant experiments, so TGA can not be used to test the thermal properties of the materials. Free volume fraction and TG are used to characterize the thermal properties of materials. This paper is based on reference [1].

[1]Du D, Tang C, Zhang J, et al. Effects of Hydrogen Sulfide on the Mechanical and Thermal Properties of Cellulose Insulation Paper: A Molecular Dynamics Simulation[J]. Materials Chemistry and Physics, 2019, 240:122153.

(7)Moreover, the references should be supplemented, as only the names of the two first authors were given

Thanks for your comments.

We’ve supplement the references.

Reviewer 2 Report

Zhang et al. report on the Molecular Dynamics study of cellulose reinforced with SiO2 nanoparticles. The simulations indicate that silane agents grafted on the SiO2 could lead to improved mechanical and thermal behavior (higher Tg and degradation temperature). This certainly provides a valuable insight that could help designing new reinforced materials.. However, the paper should depict the structures of the various silanes investigated here (KH550, KH560, KH570 and KH792) and the reinforcement mechanisms should be discussed in line with chemical structure and possible interactions in the material. How do they explain that amine-containing silanes are more efficient? The model's limitations should also be discussed as it seems unable to tell wether particles agregation is avoided, which is a key parameter as suggested from the literature. Finally, the manuscript should be checked for typos (for instance uncomplete sentences at the end of introduction or section 3.3) prior to publication.

Author Response

(1)the paper should depict the structures of the various silanes investigated here (KH550, KH560, KH570 and KH792) and the reinforcement mechanisms should be discussed in line with chemical structure and possible interactions in the material.

Thanks for your comments.

Hydroxyl groups are produced after hydrolysis of silane coupling agent. In this way, the silane coupling agent can react with the silicon hydroxyl groups on the surface of nano-SiO2, so that one end of the silane coupling agent can be connected with the surface of Nano-SiO2. The organic functional group of KH550 is amino, the organic functional group of KH560 is epoxy, the organic functional group of KH570 is acyloxy, and the organic functional group of kh792 is aminopropyl. Organic functional groups determine the binding ability between nano-SiO2 modified by silane coupling agent and cellulose. The addition of silane coupling agent enhances the compatibility between organic phase and inorganic phase to a certain extent. At the same time, the addition of silane coupling agent makes more hydrogen bonds between nano-SiO2 and cellulose, which makes the structure of cellulose more compact.

This part has been added in the revised manuscript.

(2) How do they explain that amine-containing silanes are more efficient? 

Thanks for your comments.

Amino group has strong polarity and can form more hydrogen bonds with cellulose, so it is the best to improve the properties of cellulose.

(3)The model's limitations should also be discussed as it seems unable to tell wether particles agregation is avoided, which is a key parameter as suggested from the literature.

Thanks for your comments.

It is true that the agglomeration of nanoparticles is not discussed in this paper, but the related literature [1-2] shows that grafting silane coupling agent onto nano-SiO2 can reduce the agglomeration of nanoparticles to a certain extent.

[1] He Shuting,Liu Chunbao. Surface modification of nano-SiO2, Applied Chemical Industry, 2017, 46(302):92-96. 

[2] Wu Wei, Chen Jianfeng, Qu Yixin. Influence of the kinds and structure of silicane coupling agent on polymer grafting modification of the ultrafine silicon dioxide surface, Journal of the Chinese ceramic society, 2004(05):570-575.

(4)Finally, the manuscript should be checked for typos (for instance uncomplete sentences at the end of introduction or section 3.3) prior to publication.

Thanks for your comments.

We invited a professional English native speaker from Edanz group to retouch the manuscript and check the language and spelling.

Reviewer 3 Report

An interesting paper that uses molecular dynamics simulations to investigate properties of systems consisting of cellulose and nano-SiO2 particles modified with different silane coupling agents. Such materials can improve properties of transformers, which play a crucial role in providing a reliable and efficient supply of electricity. Authors analyze a model consisting of silanized nano-SiO2 placed in the cell with the cellulose and study its mechanical properties, the energy changes during heating of the systems, thermodynamic properties, and the free volume.

1) The manuscript would benefit from some corrections in vocabulary, for example, when you say "microcosmic" do you mean "micro-scale"?

2) I list here some errors I spotted (they are too numerous, to list all of them)

line 34 Lin et al. used nano-Al2O3

line 47 Wang et al. studied

line 54 guarantee (consider assume)

line 55 tensile model modulus

line 57 cellulose modified (consider mixed)

line 58 molecular dynamics simulation technology 

line 59-60 ???? (I do not understand)

line 62 there will be a lot  many

line 66 will shrink (consider interact)

line 70 will form condensation (consider "will undergo condensation reaction")

line 71 In order to simplify the model, we assume that the other

line 81 model established (consider "used")

line 94 van der Waals action (did you mean "interactions")

line107 the modulus  values (E, G, and K) of each model  modulus

line 111 properties of the four

3) line 86-90. Can you please describe it at some length

Author Response

Thanks for your comments. Please find the reply in the attached file.

Round 2

Reviewer 1 Report

TGA can really improve the thermal properties of the reaction materials, but this paper is from the micro level, and has not carried out relevant experiments, so TGA can not be used to test the thermal properties of the materials. Free volume fraction and TG are used to characterize the thermal properties of materials. This paper is based on reference [1].

Reviewer: TGA is just the tool for thermal properties determination, not for modifying it. I understand, that TG cannot be performed as the work has more modelling and simulation character than experimental.

Reviewer 2 Report

The revised manuscript addresses previous concerns and can be published in present form.